# A Secure Traffic Police Remote Sensing Approach via a Deep Learning-Based Low-Altitude Vehicle Speed Detector through UAVs in Smart Cites: Algorithm, Implementation and Evaluation

Ata Jahangir Moshayedi [1], Atanu Shuvam Roy [2], Alireza Taravet [3], Liefa Liao [1], Jianqing Wu [1,*] and Mehdi Gheisari [4]

[1] School of Information Engineering, Jiangxi University of Science and Technology, No. 86, Hongqi Ave., Ganzhou 341000, China
[2] Department of Computer Science and Engineering, Indian Institute of Technology, Kanpur 208016, India
[3] Deimos Space, Oxford OX11 0QR, UK
[4] Department of Cognitive Computing, Institute of Computer Science and Engineering, Saveetha School of Engineering, Saveetha Institute of Medical and Technical Sciences, Tamil Nadu 602105, India
* Correspondence: jianqing.wu@jxust.edu.cn

**Abstract:** Nowadays, the unmanned aerial vehicle (UAV) has a wide application in transportation. For instance, by leveraging it, we are able to perform accurate and real-time vehicle speed detection in an IoT-based smart city. Although numerous vehicle speed estimation methods exist, most of them lack real-time detection in different situations and scenarios. To fill the gap, this paper introduces a novel low-altitude vehicle speed detector system using UAVs for remote sensing applications of smart cities, forging to increase traffic safety and security. To this aim, (1) we have found the best possible Raspberry PI's field of view (FOV) camera in indoor and outdoor scenarios by changing its height and degree. Then, (2) Mobile Net-SSD deep learning model parameters have been embedded in the PI4B processor of a physical car at different speeds. Finally, we implemented it in a real environment at the JXUST university intersection by changing the height (0.7 to 3 m) and the camera angle on the UAV. Specifically, this paper proposed an intelligent speed control system without the presence of real police that has been implemented on the edge node with the configuration of a PI4B and an Intel Neural Computing 2, along with the PI camera, which is armed with a Mobile Net-SSD deep learning model for the smart detection of vehicles and their speeds. The main purpose of this article is to propose the use of drones as a tool to detect the speeds of vehicles, especially in areas where it is not easy to access or install a fixed camera, in the context of future smart city traffic management and control. The experimental results have proven the superior performance of the proposed low-altitude UAV system rather than current studies for detecting and estimating the vehicles' speeds in highly dynamic situations and different speeds. As the results showed, our solution is highly effective on crowded roads, such as junctions near schools, hospitals, and with unsteady vehicles from the speed level point of view.

**Keywords:** vehicle detection; vehicle speed estimation; transportation; unmanned aerial vehicle; deep learning; remote sensing

## 1. Introduction.

Unmanned aerial vehicles (UAVs) are increasingly used for remote sensing (RS) applications and are a relatively new category of robots in broader (commercial) use [1]. The small size, flexible movement, and good control of UAVs [2] combined with the vision systems [3] open a new gate for measuring and capturing data remotely. Recently, RS

applications using UAVs have been used for various tasks, such as thermography timing of different thermal orthomosaics and photographs [4], boundary detection between land parcels [5], complex and irregular field shape [6], accuracy measurement [7] or validating for utilizing multi-temporal color images [8], and precision agriculture to optimize crops and facilities crops management [9]. However, visual/object tracking to locate, detect, and define objects [10] via UAV images is still challenging, even with state-of-the-art deep learning models (RESNET, Mobilenet, Efficient Net etc.) in the field of computer vision. Today, deep learning algorithms reach performance close to human experts in many applications, e.g., traffic surveillance, accident avoidance, traffic intersections [3,10], autonomous vehicles, and intelligent transport systems. In the related literature, Retina Net, FCOS, and YOLOv3 (YOLO-v3, YOLO-v3-spp and YOLO-v3-tiny) [11,12], Faster R-CNN [12–14], multi-perspective convolutional neural network R-CNN [15], and hierarchical Bayesian algorithm [16] are proposed for vehicle detection based on satellite data or stationary vision systems. Based on the authors' knowledge, there are few studies so far about the combination of UAV, visual/object tracking and speed detection. Moranduzzo et al. [3] proposed a method based on scale-invariant features transform (SIFT) for vehicle speed estimation in UAV imagery. The process begins with registering two successive images belonging to a sequence acquired by the UAV at a height of 200 m. Their results show that the proposed method generates 80.0% accuracy for car detection on the images acquired at a height of 200 m. Afifah et al. [17] estimated vehicle speed using Euclidian distance. As the first step in their processing chain, all the images are transformed with warp perspective to align them with a global coordinate. Then, they are converted to grayscale images and blurred with a gaussian function. Finally, after performing background subtraction and comparing and thresholding the images, the vehicle speed is estimated by comparing each vehicle with itself in the next frame using Euclidian distance. The accuracy of their proposed approach is 92%. However, although very low altitude (<10 m) UAV-based car speed detection systems can be very effective, a detailed analysis of them has not been performed yet. The present paper aims to demonstrate the potential of deep learning approaches in newly proposed car speed detection systems using low-altitude UAV data to help traffic safety and security in smart cities. This paper makes the following contributions:

1. A more reliable and secure solution to verify vehicle speed using drones instead of current studies is offered.
2. Providing an effective alternative method against the many reported cases of accident damages and injuries for police officers as well as fixed traffic cameras.
3. Introducing the quick speed check system, which can be used in places with limited access and in conjunction with the available speed detection system.
4. A low-altitude drone equipped with a Mobile Net-SSD method to measure vehicle speed and a network connection capability was employed to implement our approach.
5. To achieve our solution, a low-altitude drone equipped with the Mobile Net-SSD algorithm was used to detect the speed of vehicles and has the ability to connect to the traffic police.
6. Several scenarios were taken into account in various conditions, such as road intersections and settings with abrupt speed changes, to provide more accurate findings.
7. To increase the solution's speed detection accuracy, the effective cases were tested and calibrated using a drone. The system was run on Raspberry PI4B for its faster-processing speed, and memory capabilities, such as capacity and bandwidth, benefit the deep learning-based computer vision module to run smoothly.
8. In addition to our earlier contribution, which was already discussed above, a movable camera system was mounted on top of the drone, offering a number of opportunities to assess our solution at various altitudes along the X and Y axes.
9. In addition, a graphical user interface (GUI) was designed and implemented that allows us to record the environment's status, identify problematic conditions based on

speed parameters, and send alarms to the appropriate authorities was built and put into place. Additionally, we may manage the camera's status, including its movement, through the GUI.

In general, the authors of this article believe that based on the variety of tests conducted in real cases and measurements performed. The results mentioned in the following sections and the satisfactory performance of the introduced method and system can effectively solve various scientific problems. The rest of the sections are organized as follows. Section 2 introduces the system setup and methodology used for vehicle detection, tracking, and speed estimation. Section 3 provides experimental results, followed by a discussion in Section 4. Conclusions and potential paths for future work are given in Section 5.

## 2. Materials and Methods

According to the review of the past articles [18] and the investigation of the remaining cases, vehicle speed detection systems include challenges and difficulties that need to be reviewed and updated. Some of the most important limitations can be considered as follows: Structural limitations, e.g., installing cameras at a height of 3–5 m vertically on the road surface, lead to insufficient visibility on winding roads [17], the cost and maintenance limitations of fixed cameras, which cause their limited use in places with the short-term investigation times, special uses, such as schools and hospitals, which have a lower speed limit than other places, and the possibility of using various processors [19] based on the Internet of Things(IOT), considering the good accuracy and speed for vehicle speed detection and tracking mission [18]. Therefore, according to the mentioned cases, future urban and traffic management requires the use of capabilities, such as maneuvering and fast movement, of UAVs, regardless of the type and characteristics of the cameras used. The methodology of the paper is divided into four sections: system setup, which is about the proposed hardware for the speed sensing system; vehicle detection and tracking, which is the full description of the method and algorithm used for vehicle detection and tracking section of the system. The vehicle speed estimation section fully describes the VASCAR approach used for vehicle speed estimation. The system calibration section demonstrates the sets of indoor and outdoor system calibration. The whole assembly is meant to capture, read, and analyze the raw video input from the Raspberry Pi camera and give output in real-time (Figure 1).

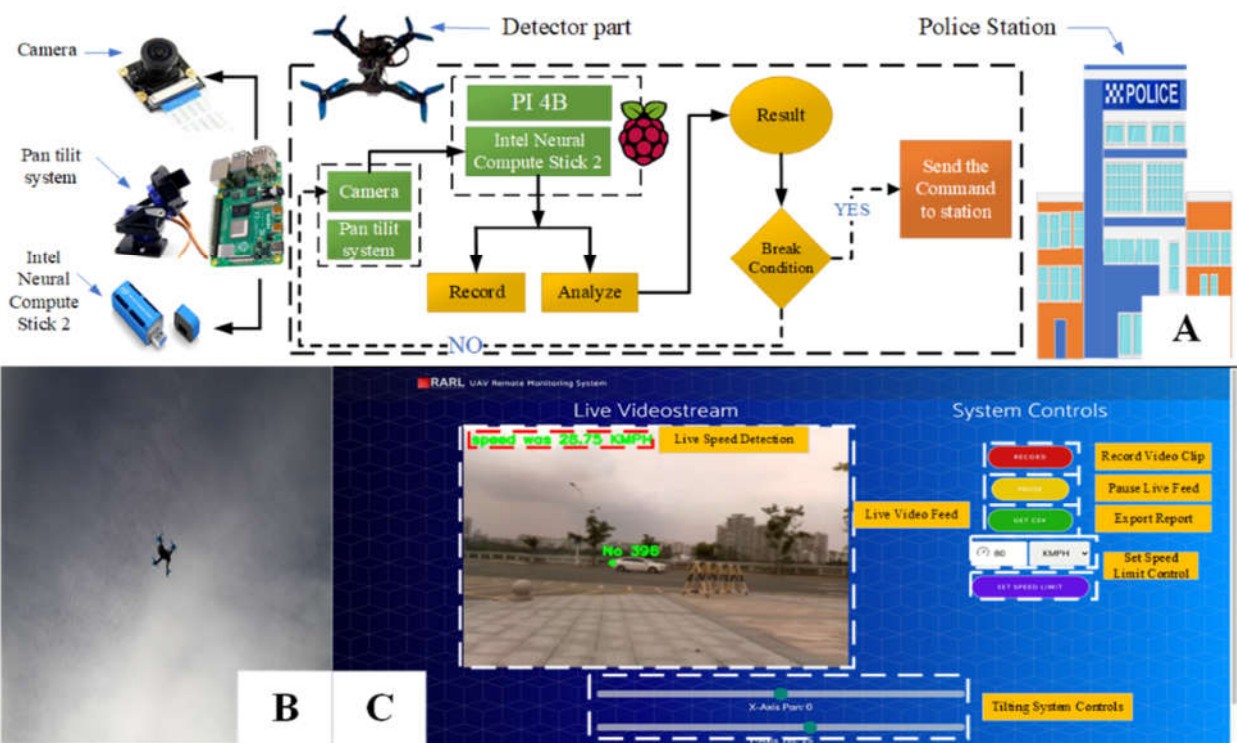

**Figure 1.** Proposed speed detection system (**A**) the total system scheme and logic, (**B**) the proposed UAV, (**C**) the system performance and designed GUI.

### 2.1. System Materials

The proposed speed sensing system consists of a UAV, CPU, and vision system, as well as an object and speed detector along with a web-based GUI (Figure 1A). The UAV (Tarot TL280H/Air 2205 2000 KV/10 × 4.5 mm) includes a control board (ArduPilot APM 2.8-3DR Robotics, USA), Motors and speed control driver (T-Motor Air 15A_China), GPS (Ublox NEO-7M u-blox, Swiss), radio controls (TX: NET-Q118G, China), radio transmitter (RG831B, 8ch 2.4 GHz, China), and battery (PULSE 2250 mAh 3S LiPo Battery, China). The vision system consists of a Raspberry Pi camera with 1.3–5 MP maximum photograph resolution (2592 × 1944), a Pan-Tilt system containing two servo motors, which can move between +90° and −90° (vertically and horizontally) to move the camera and have a controllable view, and an Intel neural compute stick 2. The Intel neural compute stick 2 is the next generation of Intel's USB plug-and-play development kit for AI and deep learning technology, powered by Intel's Movidius Myriad X Visual Processing Unit (VPU) [20]. It can be used for real-time analysis of raw video footage from cameras, which normally takes lots of time on a traditional CPU. Another reason for using this is its unique workload-specific hardware acceleration that minimizes data movement. The vision section is attached to a Raspberry Pi 4 minicomputer via a USB port. The Raspberry Pi processor and its camera (Pi camera) have been used in various Internet of Things projects and are recognized as a trustable platform [21]. Among the variety of Raspberry Pi types, the PI 4 type has been chosen as it is the latest available version and supports high speed in data processing, which is a mandatory requirement for this research. It should be mentioned as the quad needs light accessories to continue the flight of the PI camera, along with their small weight, size and processing capability, the Broadcom CPU was selected, which makes this camera unique [22]. The tilting system is connected to the Raspberry Pi GPIO connectors mounted on the UAV (Figure 1B). As Figure 1 shows, the whole system works such that after running the program inside the PI, the camera, as the first part of the sensing system, captures the video stream and feeds it into the raspberry pi computer. The

program records and analyzes the video. Then, the user inside the designed web-based GUI (Figure 1C) can monitor the live detection process, and by defining the speed value inside the GUI, any car that passes the limit receives the alarm. The design GUI (Figure 1C) has the other feature to record, pause, and export the detected car report, along with a tilting system to control the camera attached to the UAV.

### 2.2. Deep Learning Model Architecture

Inside the design, the Single Shot Multibox Detector Mobile Net (SSD-Mobile net) was used as the deep learning model. This model is designed based on (Depth wise Separable Convolutions), which are separated into different CONV layers, one for filtering and one for integrating. In other words, SSD object detection comprises two sections, extracting feature maps, and applying convolution filters to detect objects.

In this research Mobile Net-83 SSD model was selected due to features. such as good accuracy for target detection [23], small size and high speed with real-time processing performance [24], the ability to detect the object in one shot with the Multibox detector [25], along with the ability to implement on embedded system platforms. In addition, this model can detect the object [26] that is supposed to tilt the camera UAV in two axes, according to the experimental target [27]. The suggested SSD Mobile Net model is shown in Figure 2.

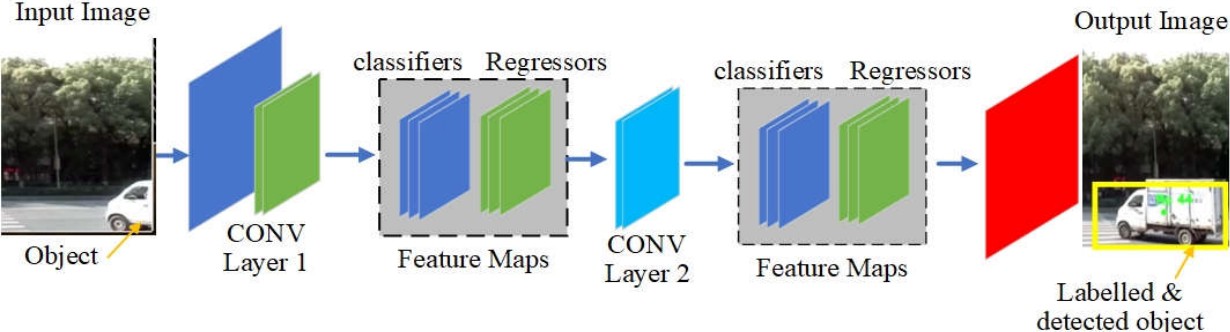

**Figure 2.** The single shot multi-box detector mobile net (SSD-mobile net) model structure.

As Figure 2 shows, the Mobile Net model aims to assign a default filter to each neural input channel for setting up the extraction of features. A (1 × 1) pointwise convolution follows next to integrate the outcome of the depthwise convolution. The batch norm comes into each of these separable layers. ReLU nonlinearity anticipates that the final (FC) layer that feeds into a SoftMax layer will be classified as having no nonlinearity. Unlike classic CNN, Mobile net's filters analyze each color channel separately before combining the three outputs into a single value. This factoring has a significant impact. In the developed method, features from the input photos are extracted using (32 and 64) filters with sizes of (5 × 5) before two max-pooling (pool size = 2). More details about the model can be found in [28].

### 2.3. Vehicle Detection and Tracking

The proposed system follows the described process in Figure 3. As shown, the main steps of work consist of initialization, which includes initializing the camera for live stream, DNN (SSD-mobile net) calling to perform in the vehicle's detection section, and speed calculation.

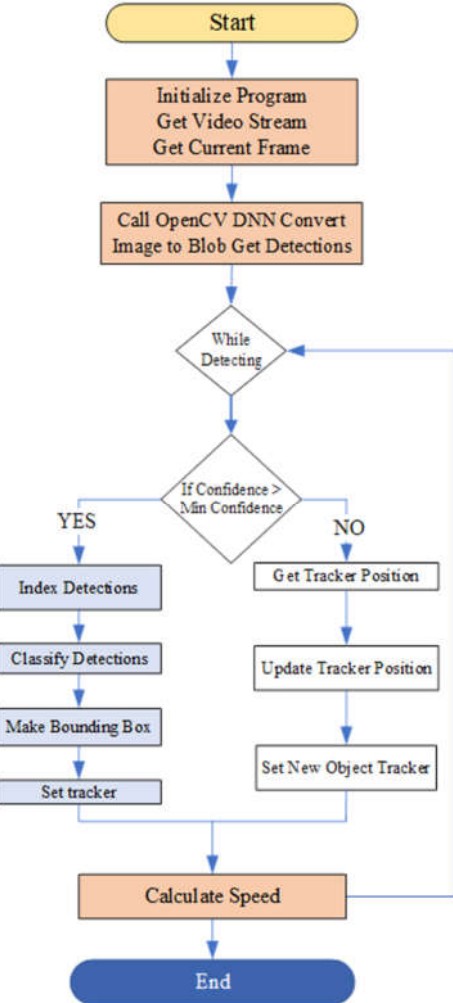

**Figure 3.** The proposed vehicle speed estimation process.

Figure 3 depicts the proposed vehicle speed estimation process which starts by receiving the video stream from the camera to calculate the speed of each tracked vehicle.

The program takes in each frame of the livestream footage and runs it through a pre-trained deep-learning model. As shown, after initialization and getting the object frame called, the OpenCV DNN converts the image blob and performs detection by confidence evaluation. The confidence evaluation means the strictness of matching each vehicle. The lower the minimum confidence specified, the less will be the detection accuracy. If the confidence of the detected vehicle is more than the minimum confidence (70%), then the vehicle will be indexed into memory and classified into a vehicle category followed by the setting of a bounding box and its position set to be tracked in the subsequent frames on the live feed. If the confidence of the tracked vehicles is less, the object's last position will be updated. Then, the new vehicle will be tracked, and the loop will keep going. The model contains classification methods for all types of vehicles, such as cars, buses, trucks, etc. Meanwhile, vehicle tracing occurs as the vehicle moves across consecutive frames and calculates and declares the speed using the formulas explained in this section.

2.3.1. Vehicle Detection Approach

The vehicle detection phase consists of a multi-step process that relies on the existing object centroid calculated in the proposed system to confirm and define the new object (Figure 4).

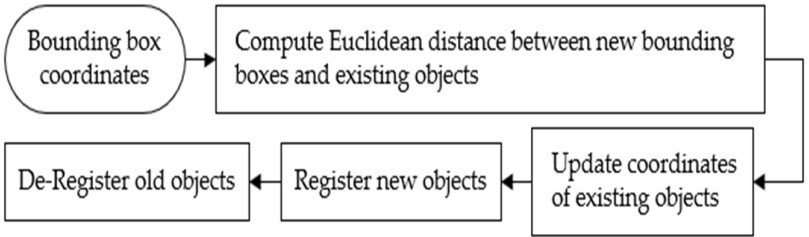

**Figure 4.** The schematic view of the vehicle detection and tracking algorithm.

The first task in the vehicle detection and tracking phase is to identify the objects and build bounding boxes around them. For this purpose, all three bands (Red, Green, and Blue) of each frame of the video are normalized using the following equations (Equation (1)):

$$\text{NormBand} = \frac{(\text{Band} - \mu_{\text{BAND}})}{\sigma} \tag{1}$$

where Band is band values, and $\mu_{\text{BAND}}$ is the mean of each red, green, and blue Band, respectively, $\sigma$ is the scaling factor for normalization. After normalization, Mobile Net-SSD is used for detecting vehicles. Mobile Net-SSD is a Single-Shot multi-box Detection (SSD) network intended to perform object detection. The vehicle detection and tracking system loops over all detected vehicles, add a bounding box around them and then calculates the centroid of the boxes. After the bounding box coordinates are extracted, the Euclidean distances between the new and old bounding boxes are calculated. Each video frame can have a different position of the previously tracked object, leading to different boxes assigned to the same object in different frames. To avoid this, the distance between the new object in the next frame and the old object in the last frame is calculated using the following equation (Equation (2)):

$$d(x,y) = \sqrt{\sum_{i=1}^{n}(y_i - x_i)^2} \tag{2}$$

where $x$ and $y$ are $x$ and $y$ coordinates, respectively, and $i$ is the instance. For instance, suppose that the last object detected in the frame is Ft, and the newly detected object is Ft + 1 (where it is the current frame). If the old and newly detected objects refer to the same object, then the distance between Ft and Ft + 1 will be less than a new object's length. Hence, the algorithm associates newly detected objects with previous ones' consecutive frames and updates their position. In the last step, tracked vehicles that have not been visible in 4 frames are removed and not tracked anymore.

2.3.2. Vehicle Speed Estimation Using an Improved VASCAR approach

This study's vehicle speed estimation model is based on an improved Visual Average Speed Computer and Recorder (VASCAR) method [29]. This method is based on timing and the known distance between two fixed points on the road, as shown in Equation (3). When a car passes the first reference point, the detector triggers the timer and captures the time until the car passes the last point to calculate velocity.

$$\text{Vehicle speed} = \frac{\text{distance between Points (A} - \text{B)}}{\text{time}} \tag{3}$$

In the case of human operation, this method is severely limited by human error and delayed reaction. This research considers four instead of only two points and automatically calculates the velocity between the reference points. The standard VASCAR is

calculated based on three points, while in the improved VASCAR, four measurements have been considered to calculate velocity. The measurement process is shown in Figure 5.

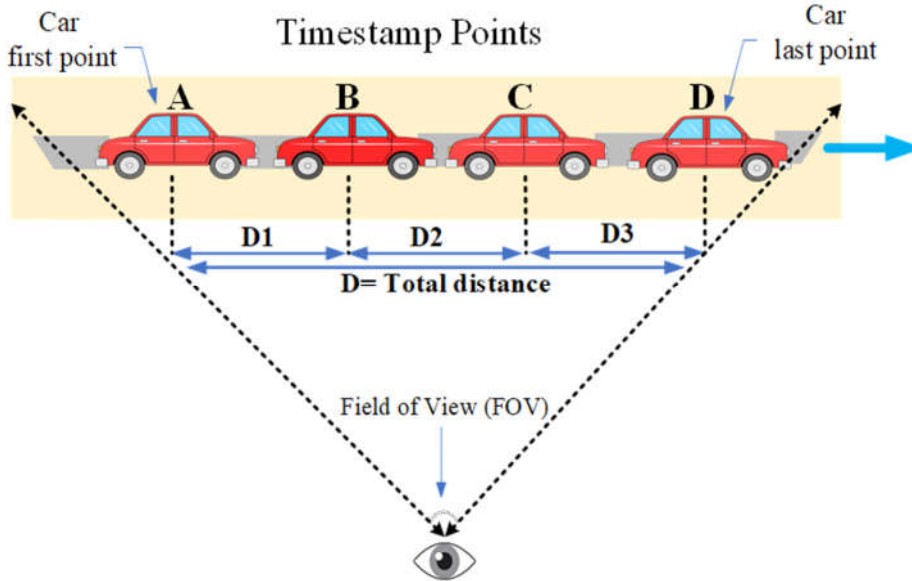

**Figure 5.** Schematic view of the VASCAR approach in different location of A to D.

As shown in Figure 5, the car was detected at the specified point with a known distance. The system calculates the field of view based on the distance from the road it observes. The speed estimation system divides the whole frame into four points, considering the frame points in the video to track the object. When a car passes reference point A, the detector triggers the timer and captures the time until the car passes point D. This automatic approach overcomes human error and delayed reaction that might occur in the case of human operation. The calculations are shown in Equation (4).

$$\text{Average Speed (km/h)} = \left(\frac{D1}{t1} + \frac{D2}{t2} + \frac{D3}{t3} / 3\right) \times 3.6 \tag{4}$$

where $D_1$ is the distance between points A–B in meters at $t_1$ second, the distance $D_2$ between points B–C as $t_2$ second, and the points C–D distance as $D_3$ meter with $t_3$ second. As Figure 5 shows, tracking an object in four points results in three distances included in the relationship to calculate the average. In the next step, the system calculates the field of view (FOV) based on the distance from the object. From the system point of view, at the same time, new vehicles are constantly being tracked and registered, and old objects are being deregistered. Meanwhile, the improved VASCAR algorithm is applied to each tracked vehicle. In the second phase, the system initializes the estimated speed list, loops over all the pairs of points, and estimates the speed of each object. Then, it calculates the number of pixels between centroids in points and converts it to real-world distance(meters) as the pixel per meter (PPM) in Equation (5) and calculates the vehicle's average speed.

$$\text{PPM} = \frac{\text{Distance Constant}}{\text{Frame Width}} \tag{5}$$

Equation (6) calculates the pixel spacing difference between the vehicle passing through each point for each point pair (e.g. A and B).

$$P_{AB} = |\, Coords_B - Coords_A| \tag{6}$$

where Coords$_B$ and Coords$_A$ are centers of points A and B, respectively. Finally, Equation (7) is used to calculate the distance in meters (d$_{AB}$).

$$d_{AB} = P_{AB} \times PPM \tag{7}$$

Four timestamps are stored when the vehicle passes the video frame columns to calculate the average speed. The average speed is calculated using the following equation (Equation (8)).

$$Average\ Speed = \frac{\dfrac{\Delta t_{AB}}{d_{AB}} + \dfrac{\Delta t_{BC}}{d_{BC}} + \dfrac{\Delta t_{CD}}{d_{CD}}}{3} \tag{8}$$

where $\Delta t_{AB}$ is the timestamp between points a and b, $\Delta t_{BC}$ is the timestamp between points b and c, and $\Delta t_{CB}$ is the timestamp between points c and d in second. In this paper, based on mentioned equations, the improved VASCAR algorithm is implemented as a Python-based program in OpenCV, and the DNN is built. The program acts to detect the object as the vehicle detection and tracking, and then it tracks and estimates the speed as the vehicle speed estimation.

*2.4. System Calibration*

Some points should be considered while using a speed detector system camera. Pixel per meter ratio is one of the parameters which should be calibrated. It represents the slice of road covered by each pixel and is relative to the square distance from the camera. Notable factors include (a) camera height above ground (needs to be in the range of: distant (≥5 m) or close (<5 m)), and (b) camera location (should be on the side of the road) [30]. Camera focal length is another important parameter mainly related to camera height, the length of the road segment, and the number of lanes covered by the field of view. In most related research, this number is reported as ≤25 mm [30] to cover multiple lanes and a large road stretch. Furthermore, system calibrations are carried out to better evaluate the proposed system and algorithms. This step is essential to tune system parameters and have some pre-analysis of the system. The calibration process contains the indoor and outdoor calibration from the tested FOV of the used camera to the program parameter and the final speed detection calibrated formula, which is described as follows:

2.4.1. Indoor System Calibration

Indoor system calibration extracts the camera's field of view (FOV). FOV is the maximum sample area that a camera can image. It is calculated in three steps (see Figure 6): horizontal, vertical, and diagonal. A diagonal field of view (DFOV) designates the diagonal dimensions of the measurement area in the object plane; a horizontal field of view (HFOV) defines the horizontal dimensions of the object plane, and a vertical field of view (VFOV) represents the vertical dimensions of the measurement area in the object plane [31].

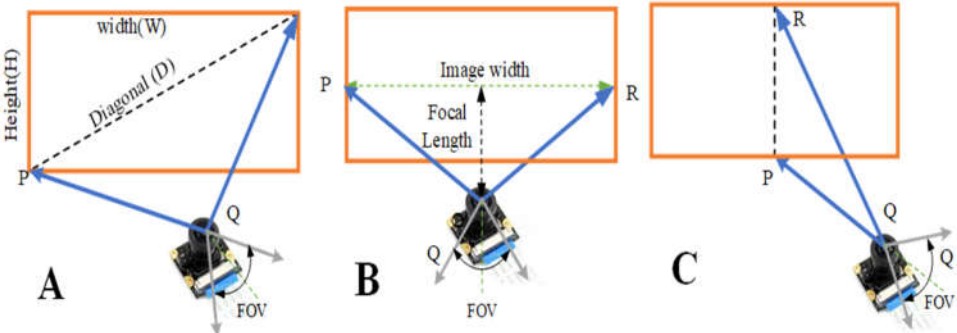

**Figure 6.** Different camera FOVs measurement situation, diagonal lengths (**A**), horizontal (**B**), and vertical (**C**).

As shown in Figure 6, to extract the maximum Pi camera FOV, the camera has been moved on along with points P and Q. Then, using Equation (9), the camera horizontal FOV value can be defined by:

$$\text{FOV} = 2 \tan^{-1}\left(\frac{1}{2}\frac{\text{image width}}{\text{focal length}}\right) \tag{9}$$

FOV extraction includes the following steps based on the setup shown in Figure 6:

- Step 1: Calculates the horizontal, vertical, and diagonal FOV from a fixed distance from the object at different heights.
- Step 2: FOV extraction of the camera in different tilted positions on the x-axis to check any orientation effects on the Pi's FOV.
- Step 3: FOV extraction of the camera in different tilted directions of the y-axis while the camera is 20 cm from the object. The same principle of the trapezoid as the last test applies, but in this case, the trapezoidal frame appears along the vertical side.
- Step 4: Finally, the camera performance was studied with a random test for any height and degree to calculate the FOV.

As Figure 6A shows, the camera system was put on top of the tilt systems to move the camera on various x- and y-axes. It should be mentioned that for each tilt, the trapezoidal area of the image gets changed (Figure 6B–D) [32]. Then, to calculate the FOV, the median of the trapezoidal (MT) image area is recalculated using the Equation (10).

$$\text{MT} = (L1 + L2)/2 \tag{10}$$

where L1 and L2 are the lengths of base 1 and 2, respectively, and MT (median of the trapezoid) is the image width used to calculate the FOV of the respective tilt angle.

Figure 7 shows the tilting system assembled with UAV and controlled with Raspberry Pi. As shown in Figure 7A, the assembled servo motor can move in ±90 in X and Y directions. Figure 7B shows that changing the baseline of L1 and L2 can cause the three parallel situations. Figure 7C: horizontally angled. Figure 7D: vertically tilted, which can affect the acquired image by the camera. It should be mentioned that for the Outdoor test, the same assembly without a stand is installed on the top of the UAV.

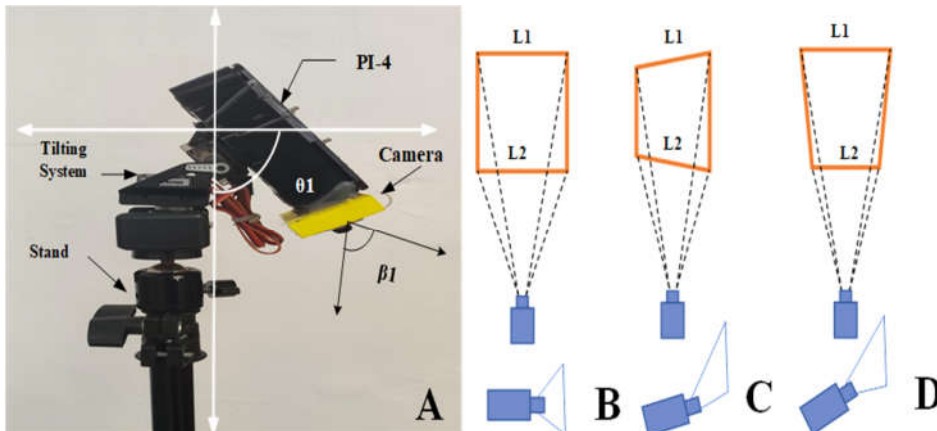

**Figure 7.** Indoor setup for the tilt system with the servo bed. (**A**) tilt system assembly, consisting of two servo motors capable of moving in the x and y axis along with raspberry PI 4B, pi camera. (**B**) the image and camera situation in a different scenario, parallel to the surface, (**C**) horizontally angled, (**D**) vertically tilted, L1: base 1 and L2: based 2.

#### 2.4.2. Outdoor System Calibration

The whole setup was brought outside the lab and tested to find the best distance for object detection. The maximum distance was taken as the best distance because it would allow a single vehicle to traverse further in the frame. In each test, the FOV was recorded for the corresponding distance with the help of a measuring tape. The next step in outdoor system calibration is system parameter calibration. The max disappears, and distance and frame width should be correctly assigned to have the best system performance for different FOVs.

## 3. Experimentation and Results

### 3.1. Indoor System Calibration

The first indoor system calibration step was to find horizontal, vertical, and diagonal FOV from a fixed distance from the object at different heights. The experiment was conducted based on the setups shown in Figure 8.

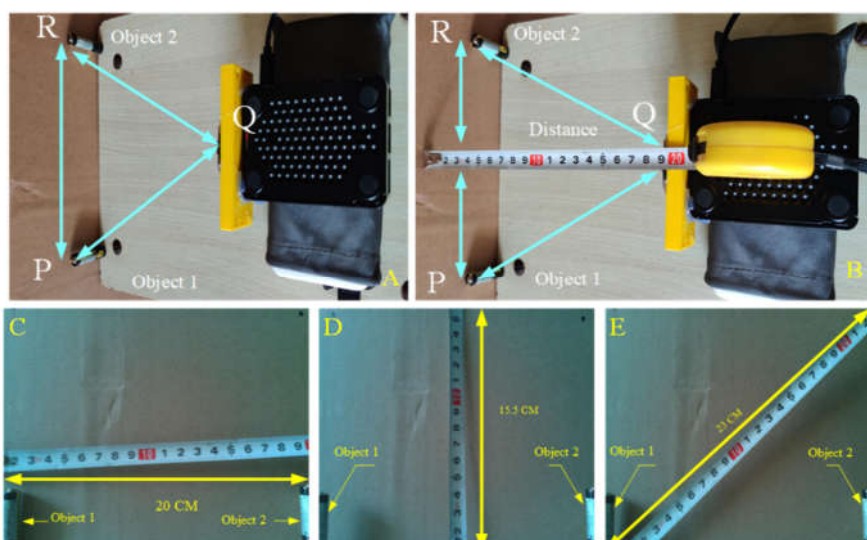

**Figure 8.** Indoor setup for horizontal, vertical, and diagonal FOV extraction from a fixed distance. (**A**): Distance between fixed object and camera, (**B**): Real Horizontal Length measurement, (**C**):

Horizontal lengths inside the obtained image, (**D**): vertical lengths inside the obtained image, (**E**): diagonal lengths inside the obtained image.

To test the camera indoors FOV parameter, as the first step based on Figure 7, the Raspberry Pi with the camera mounted was set on a surface with a fixed object distance of 20 cm and varying height (r) from 0.25 to 3.5 m. The obtained results are shown in Table 1.

**Table 1.** Indoor FOV measurement with fixed distance (20 cm) and varying heights, Height (H), Horizontal Length (HL), Horizontal Fov (H_FOC), Vertical Length (VL), Vertical Fov (V_FOV), Diagonal Length (DL), Diagonal Fov (D_FOV).

| H (m) | HL (cm) | H_FOC (Degree) | VL (cm) | V_FOV (Degree) | DL (cm) | D_FOV (Degree) |
|---|---|---|---|---|---|---|
| 0.25 | 19.5 | 51.9784 | 15.5 | 42.3626 | 23 | 59.7978 |
| 0.5 | 19.6 | 52.2097 | 15.4 | 42.1134 | 23.2 | 60.2274 |
| 0.75 | 20 | 53.1301 | 14.9 | 40.8607 | 23.1 | 60.0128 |
| 1 | 20 | 53.1301 | 15 | 41.1120 | 22.8 | 59.3662 |
| 1.25 | 20.3 | 53.8155 | 15.2 | 41.6135 | 23.5 | 60.8684 |
| 1.5 | 19.8 | 52.6708 | 15.5 | 42.3626 | 23 | 59.7978 |
| 1.75 | 19.6 | 52.2097 | 15.3 | 41.8637 | 23 | 59.7978 |
| 2 | 20.3 | 53.8155 | 15.2 | 41.6135 | 23 | 59.7978 |
| 2.25 | 20.3 | 53.8155 | 15.2 | 41.6135 | 22.9 | 59.5822 |
| 2.5 | 20.1 | 53.3590 | 15.1 | 41.3630 | 23 | 59.7978 |
| 2.75 | 19.5 | 51.9784 | 15.1 | 41.3630 | 23 | 59.7978 |
| 3 | 20 | 53.1301 | 15.1 | 41.3630 | 23.1 | 60.0128 |
| 3.25 | 20.1 | 53.3590 | 15 | 41.1120 | 23 | 59.7978 |
| 3.5 | 19.9 | 52.9006 | 15 | 41.1120 | 23 | 59.7978 |

Table 1 concludes that the results of every test actually can coincide with the design FOV angles from the specifications (H_FOV: 53.50 +/− 0.13 degrees, V_FOV: 41.41 +/− 0.11 degrees) and considering that the 2-degree measurement error differential due to human error and the height does not affect the FOV by a huge margin, be it horizontal, vertical, or diagonal. It should be noted that because of the angle change, the image received or captured may seem rectangular on screen, detecting a trapezoidal frame. In the next step, the FOV variations in the fixed distance (20 cm) with different camera angles in the x-direction from −90 to +90 degrees are presented (Table 2).

**Table 2.** Indoor FOV(Filed Of View) measurement over changing the camera angle from −90 to +90 angle of view on the x-axis (AOV_X**)**, Distance From Lens (DL), Horizontal Length (HL), Horizontal FOV (H_FOV), Vertical Length of Base 1 (VL1), VERTICAL LENGTH of BASE 2 (VL 2), Vertical Median of Trapezoidal Frame(VMT**),**Vertical FOV (V_FOV),Diagonal Length(DL), Diagonal FOV (D_FOV).

| AOV_X (Degrees) | DL (cm) | H_L (Degree) | H_FOV (Degree) | VL1 (cm) | VL2 (cm) | VMT (cm) | V_FOV (Degree) | DI_L (cm) | D_FOV (Degree) |
|---|---|---|---|---|---|---|---|---|---|
| 90 75 60 | Obj. out of scope | Null | 0 | Null | Null | Null | 0 | Null | 0 |
| 45 | 22.5 | 23 | 54.1441 | 15 | 18 | 16.5 | 40.272 | 35 | 75.7499 |
| 30 | 21.5 | 21 | 52.0591 | 15 | 17 | 16 | 40.819 | 32 | 73.3122 |
| 15 | 21 | 21 | 53.1301 | 15 | 15 | 15 | 39.307 | 30 | 71.0753 |
| 0 | 20 | 20 | 53.1301 | 15 | 15 | 15 | 41.112 | 23 | 59.7978 |
| −15 | 21 | 21 | 53.1301 | 15 | 16 | 15.5 | 40.512 | 30 | 71.0753 |
| −30 | 21.5 | 21 | 52.0591 | 15 | 17 | 16 | 40.819 | 32 | 73.3122 |

| −45 | 22.5 | 21 | 50.0337 | 15 | 18 | 16.5 | 40.272 | 35 | 75.7499 |
|---|---|---|---|---|---|---|---|---|---|
| −60 −75 −90 | Obj. out of scope | Null | 0 | Null | Null | Null | 0 | Null | 0 |

As Table 2 shows, for the degrees 60 or above, and in the same case for below −60, when the device was turned to that specific degree, the object's frame was out of the image frame. This test shows that the camera movement on the tillite system in the x direction should be between ±60 degrees. For the third test, the camera gradually tilted along the y-axis while the object was at 20 cm in a constant place. The same principle of the trapezoid as in the last test applies. In this case, the trapezoidal frame appears along the vertical side. It should be noted that due to the y-axis movement of the camera, only one base (base 2) of the image will be changed (Table 3).

**Table 3.** Indoor FOV measurement with device position from the object (cm) and changing the camera in the y direction, Distance From Lens (DL), Horizontal Length (HL), Horizontal FOV (H_FOV), the Vertical Length of base 1 (VL1), the vertical length of base 2 (VL2), Vertical Median of Trapezoidal Frame (VMT), Vertical FOV (V_FOV), Diagonal Length (DI_L), Diagonal FOV (D_FOV), object out shown as OUT.

| AOV_Y (Degrees) | DL (cm) | VL2 (cm) | VMT (cm) | H_FOV (Degree) | VL1 (cm) | V_FOV (Degrees) | DI_L (cm) | D_FOV (Deg) |
|---|---|---|---|---|---|---|---|---|
| 90 75 | Obj. out of scope | Null | Null | 0 | Null | 0 | Null | 0 |
| 60 | 40 | 42 | 31 | 42.3626 | 16.1 | 29.113 | 39 | 64.342 |
| 45 | 34 | 36.5 | 28.25 | 45.1200 | 15.8 | 31.246 | 35 | 63.553 |
| 30 | 27 | 32 | 26 | 51.4199 | 15.5 | 33.196 | 32 | 63.215 |
| 15 | 24 | 27 | 23.5 | 52.1711 | 15.2 | 35.842 | 30 | 65.100 |
| 0 | 20 | 20 | 20 | 53.1301 | 15 | 41.112 | 23 | 59.797 |
| −15 | 24 | 26.5 | 23.25 | 51.6887 | 15 | 35.757 | 30 | 65.657 |
| −30 | 27 | 32 | 26 | 51.4199 | 15.4 | 32.993 | 32 | 63.215 |
| −45 | 34 | 36 | 28 | 44.7602 | 15.6 | 31.132 | 35 | 64.010 |
| −60 | 40 | 41.5 | 30.75 | 42.0510 | 15.9 | 28.991 | 39 | 64.761 |
| −75 −90 | Obj. out of scope | Null | Null | 0 | Null | 0 | Null | 0 |

Table 3 shows that the system has a limitation of ±75-degree rotation in the direction of the object inside the frame, so the direction of movement should rotate within ±75 degrees. As the result shows, the degree between 75 and 90 in the Y direction with positive and negative values causes the camera disability to record an image, so the system loses its performance to track the object.

*3.2. Outdoor System Calibration*

The outdoor system calibration starts with checking the Pi camera FOV. In order to estimate vehicle speed accurately, the vehicle must be seen traversing from the initial point of the frame to the very end. Table 4 shows the various FOVs measured from various distances from 300 up to 1600 cm horizontally perpendicular to the road. All the outdoor system calibrations are conducted at a laminar wind speed.

**Table 4.** Outdoor Field Of View (FOV) measurement for various distances from the road, Average Horizontal Field Of View (AVG H_ FOV).

| Distance From Road (cm) | FOV (cm) | FOV (Degree) | AVG H_FOV (Degree) | FOV (Degree) |
|---|---|---|---|---|
| 300 | 335 | 48.2 | | |
| 350 | 385 | 48.8 | | |
| 400 | 430 | 48.8 | | |
| 450 | 465 | 51.6 | | |
| 500 | 515 | 51.7 | 51.12 | 53.1 |
| 550 | 570 | 51.5 | | |
| 600 | 570 | 53.5 | | |
| 650 | 650 | 53.1 | | |
| 700 | 770 | 51.53 | | |
| 1600 | 1500 | 51.53 | | |

In each test, the FOV was recorded for the corresponding distance with the help of a measuring tape. As Table 5 shows, the data obtained for the tested FOV are very close to the value declared by the camera manufacturer (53 degrees), which indicates the accuracy of the calculations and experiments. Finally, the last distance of 16 meters was set as the fixed distance for future tests due to the clear image visibility at this setup. At this distance, the FOV is 15 m.

**Table 5.** Final calibration of the system parameters.

| Component | Description | Value (Unit) |
|---|---|---|
| max_disappear | Maximum consecutive frames for an object to be allowed to pass before deregistering it | 15 frames |
| max_distance | Maximum distance between centroid to associate an object | 1.75 m |
| track_object | Number of frames to track for object | 4 frames |
| confidence | Minimum confidence or probability of detection | 0.4 |
| frame_width | Frame width in pixels | 480 pixels |
| speed_estimation_zone | Speed estimation columns | 4 (A, B, C, D) |
| distance | Distance from road to camera | 16 m |
| speed_limit | Speed limit | To Be Set |

*3.3. System Software Parameters Calibration*

As already mentioned, max disappears, max distance, track object, and frame width must be determined correctly for better system performance. The parameters have been calibrated through trial and outdoor experiments with the values, as presented in Table 5.

As shown in Table 5, the parameters used for max disappear and max distances are 15 and 1.75 m. The frame number for object tracking is set to 4. The confidence was chosen to be 0.4 (i.e., minimum detection percentage of 40%). This allows some flexibility because the model will also track objects it is less confident about. Since vehicles come in all shapes and sizes, this can increase robustness in real-world use. Frame width is set to 480 as both 60 fps and 90 fps footage are supported at this resolution. Moreover, it reduces computational needs and speeds up the process due to faster calculations on smaller images.

*3.4. Vehicle Speed Estimation Optimization*

A real vehicle test shows the difference between real speed and the system's estimated speed in two road directions and this difference is higher due to the greater distance of the quad from the road. To overcome this issue, as the last step, the final calibrated formula extracted (Equations (10) and (11)) with the help of curve fitting tools in MATLAB based on the following process is applied to the outcomes of the vehicle speed detection

model. For this aim, the UAV was placed at a height of 3 m in the fixed point test, and a vehicle with a speed range of 10–70 km/h was used to test the system, considering the maximum speed limit of the city traffic police. A car with a driver and one of the researchers as an assistant driver was used. Before the vehicle starts to move, the driver is informed about the target speed then the driver brings the car's speed to the desired speed at the UAV location. Then, when passing the drone's position, along with checking vehicle identification, the UAV speed reading and the vehicle speed from the odometer are read and recorded by the driver's assistant. It should be noted that speed comparison is based on the two-way remark as direction 1, the path which has the minimum distance with the UAV and the vehicle is moving towards the UAV, and direction 2, going away from the UAV with the maximum distance and located on the other side of the street. The test is repeated five times, and the average is considered the final detected speed. Table 6 shows the system performance in real vehicle tests for the mentioned speed range.

**Table 6.** The real car speed and detected speed with the proposed system over different car directions (Direction 1: near side and Direction 2: faraway side).

| Speed (km/h) | Direction 1 (y1) | Difference | Direction 2 (y2) | Difference |
|---|---|---|---|---|
| 10 | 7.76 | 2.24 | 6.96 | 3.04 |
| 20 | 15.62 | 4.38 | 15.29 | 4.71 |
| 30 | 23.18 | 6.82 | 24.91 | 5.09 |
| 40 | 33.729 | 6.271 | 54.26 | −14.26 |
| 50 | 43.1 | 6.9 | 32.83 | 17.17 |
| 60 | 73.08 | −13.08 | 73.32 | −13.32 |
| 70 | 83.22 | −13.22 | 80.5 | −10.5 |

The result shows that when the UAV is in the car's direction side, the detected speeds are nearer than the cases with more than 16 m (Figure 9).

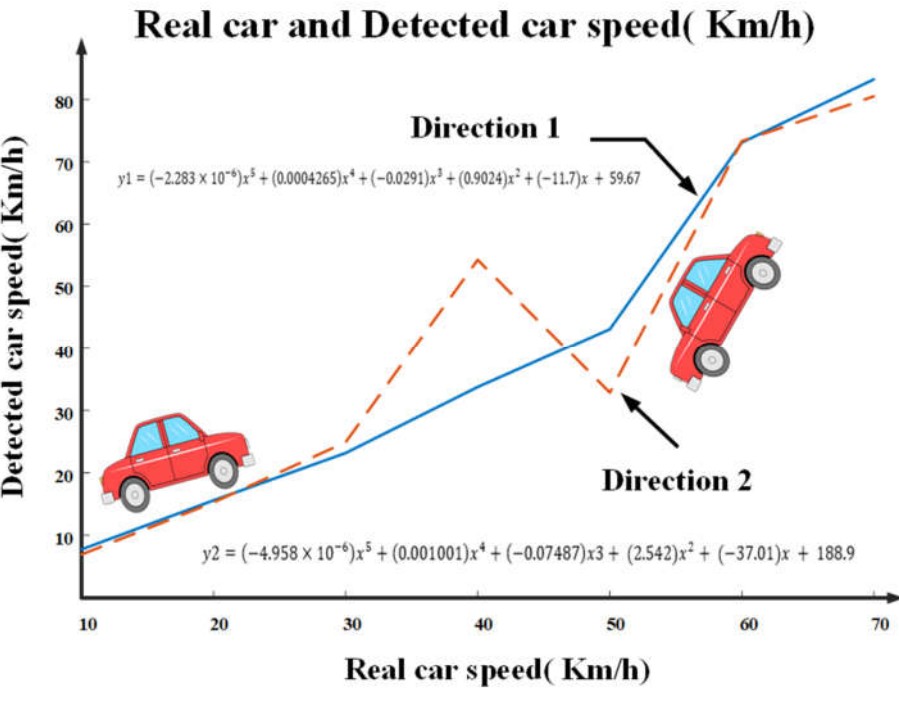

$$y1 = (-2.283 \times 10^{-6})x^5 + (0.0004265)x^4 + (-0.0291)x^3 + (0.9024)x^2 + (-11.7)x + 59.67$$

$$y2 = (-4.958 \times 10^{-6})x^5 + (0.001001)x^4 + (-0.07487)x3 + (2.542)x^2 + (-37.01)x + 188.9$$

**Figure 9.** The measured and real vehicle speed differences in two directions.

Figure 9 shows that the car has a different curve in each direction. Then, to have a better-speed estimation, the results from Table 7 were formulated with the help of the curve fitting tool in MATLAB software, and the extracted curve is shown in Equations (11) and (12).

$$y1 = (-2.283 \times 10^{-6})x^5 + (0.0004265)x^4 + (-0.0291)x^3 + (0.9024)x^2 + (-11.7)x + 59.67 \tag{11}$$

$$y2 = (-4.958 \times 10^{-6})x^5 + (0.001001)x^4 + (-0.07487)x3 + (2.542)x^2 + (-37.01)x + 188.9 \tag{12}$$

As per as analysis in MATLAB with the mentioned coefficient in Equations (10) and (11), the best R2 for direction 1 is 0.9968 and for direction 2 is 0.9033.

**Table 7.** Vehicle Detection Accuracy on the UAV Height Variation.

| UAV Height (m) | Number of Vehicles on Street | Number of Detected Vehicles | Detected Car (Error %) |
|---|---|---|---|
| 0.7 | 32 | 42 | 31.25 |
| 1.0 | 19 | 24 | 26.32 |
| 1.25 | 25 | 31 | 24.00 |
| 1.50 | 32 | 38 | 18.75 |
| 1.75 | 34 | 40 | 17.65 |
| 2.50 | 34 | 38 | 11.76 |
| 3.0 | 34 | 38 | 11.76 |

### 3.5. Real Vehicle Test

The real vehicle test includes experiments regarding changes in UAV height from 0.7 to 3 m with the fixed location, changing the UAV degree with X-axis from −15 to +15, changing the camera degree with X-axis from −90 to +90, changing the camera degree with Y-axis from −90 to +90. Finally, a random test with dynamic height and degree was conducted (Figure 10). A vehicle was driven along a specific road (JXUST university golden campus, Ganzhou, China) with minimum traffic.

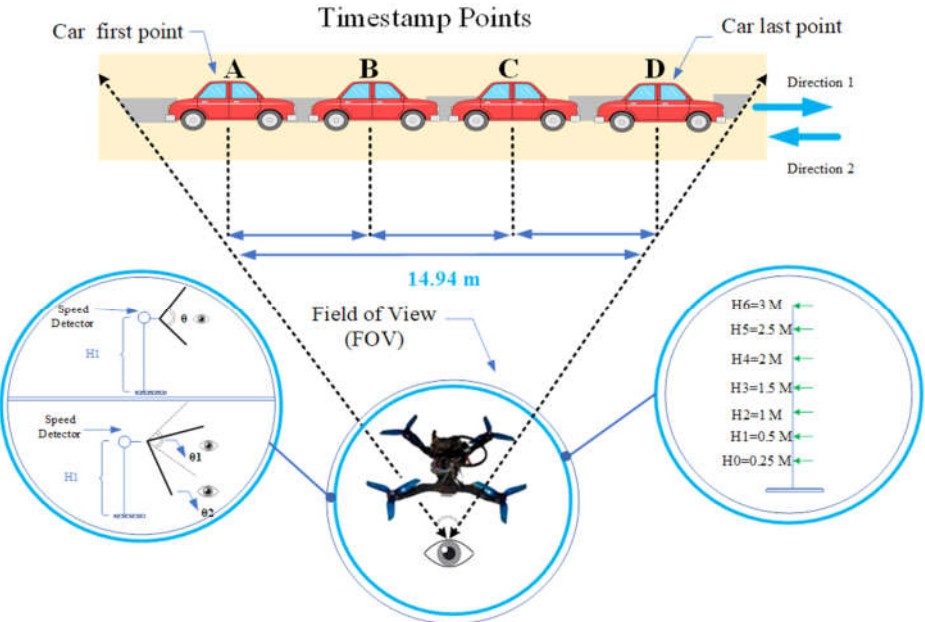

**Figure 10.** The schematic view of real vehicle tests with various location of A to D.

### 3.6. Vehicle Detection and Tracking

In the experiments, the car driver is asked to set a specific speed, so the real speed of the vehicle can be compared to the one obtained by the proposed system. In the first stage, the UAV was flown from 0.7 m height to 3 m. The maximum height of 3 m is assigned based on security issues. At each height, the estimated car speed was logged. The vehicle detection results on both roadsides are shown in Table 8.

In the next step, the car detection algorithm is tested when changing the UAV angle to the road with the x-axis from −15° to +15° at a UAV height of 3 m. The vehicle detection results on both roadsides are shown in Table 8.

**Table 8.** Vehicle detection accuracy on the UAV X-axis degree variation.

| X-Axis Change (Degree) | Number of Vehicles on Street | Number of Detected Vehicles | Detected Car (Error %) |
|---|---|---|---|
| −15 | 39 | 36 | 7.69 |
| +15 | 26 | 30 | 15.38 |

The previous test is repeated for the changes in the Y-axis degree with the UAV height of 3 m. The results show that the UAV can monitor the road between +15 and −30 in the Y direction Table 9.

**Table 9.** Vehicle detection accuracy on the UAV y-axis degree variation.

| Y-Axis Change (Degree) | Number of Vehicles on Street | Number of Detected Vehicles | Detected Car (Error %) |
|---|---|---|---|
| −30 | 22 | 28 | 27.27 |
| −15 | 18 | 21 | 16.67 |
| +15 | 21 | 25 | 19.05 |
| +30 | The road is out of range | | |

The last test is the system performance in a random situation with dynamic UAV height and a camera degree. This test in each step is repeated 5 times randomly, and the average is considered the final value is shown in Table 10.

**Table 10.** Vehicle detection accuracy on the system random test.

| Random Parameter | Vehicles on Street (KM/h) | Detected Vehicles (KM/h) | Detection Error % |
|---|---|---|---|
| UAV height 1.5 m | 49 | 58 | 18.37 |
| UAV height 1.0 m | 34 | 44 | 29.41 |
| UAV height 2.0 m | 42 | 46 | 9.52 |
| x-axis −15 degree | 36 | 39 | 8.33 |

### 4. Discussion

This research investigated low-amplitude UAV applications for car speed detection using Mobile Net-SSD models. Different indoor and outdoor tests were conducted to explore the camera's ability and calibrate the hardware. The special advantages of this design in low-altitude flights include battery energy consumption, reduced UAV charging speed, longer flight length, and the ability to develop a design to track vehicle speed at dynamic speed points, such as intersections, Joins of roads, traffic lights, etc. Another reason to check the vehicle's speed at a low height is to use a camera with fewer pixels and increase the UAV operating hours, as well as not interfere with the wind speed in the performance and control of the quad. In addition, this solution can be used in intersections and schools, kindergartens, and hospitals where vehicles generally accelerate at once. The system setup calibration demonstrated that a centroid tracking algorithm used for vehicle detection and tracking requires the camera to be perpendicular to the road to prevent obstacles. It seems that using any value for max disappear more than 15 with the given resolution of 640 × 480 would render speed detection impossible in most cases. Max distance

higher than 175 cm could cause the centroid to disassociate from the object and fail to detect the vehicles.

Furthermore, it has been observed that heat substantially reduces the performance of the system. It affects the Intel neural compute stick (system shutdown due to the heat). The results show that vehicle detection error reduces when the UAV height increases. This means that by increasing the UAV height, vehicle overlapping decreases, and sudden speed changes in vehicles can be detected easier. A system with a UAV positioned at lower heights leads to a loss in the system's tracking process. Experimental results of vehicle speed detection show that the minimum speed detection error obtained from a setup for the camera has a −15-degree angle to the X-axis, and the maximum error is a −30-degree angle. Another parameter which influences the system's accuracy is the distance from the road. As the vehicle moves further, the accuracy of vehicle detection reduces. This is especially critical for highways and wide roads because vehicles will not always drive along the road edge [33,34]. High-speed vehicles are more likely to avoid detection by the UAV speed detector. In addition, the speed of the vehicle is also one of the factors influencing the accuracy of the system, since vehicles do not always move at a constant speed. Sometimes, they may reduce speed to avoid a collision or react suddenly to a traffic situation. Moreover, vehicles are partly obscuring each other in a 2D frame due to their variable speeds. They sometimes stop in front of the camera and block other vehicles behind them. Final observations from the data show that any speed above 50 km/h reduces the system's performance. The UAV used in this project can also be autonomously controlled using modern trajectory planning algorithms, such as the evolutionary trajectory planning algorithm (ETPA), which is based on deep learning [35,36]. Besides the system can be equipped with pedestrian detection so that the UAV maintains its path avoiding pedestrians [37,38]. The mentioned improvement and utilization of modern technologies in making cities smart, especially in the transportation department, increases various parameters, such as the economy, environment, and infrastructure, from 10% to 30% [39]. Moreover, the data (image, video, graphics etc.) from these applications can help find behavior and support predictions using big data learning and discovery techniques [40].

## 5. Conclusions and Future Work

In this research paper, we have proposed a novel solution using extended UAVs, an application of remote sensing, to improve IoT-based smart city services. To show its superior performance, several different indoor and outdoor scenarios have been evaluated through implementation. The superior performance mainly stems from the accurate calibration of the UAV's camera, obtained in one of the implemented scenarios, supporting accurate vehicle speed detection. The results show that vehicle detection error reduces when the UAV height increases. Specifically, by increasing the UAV height, vehicles' separations are decreased, facilitating the easier detection of sudden speed changes in vehicles. Even in the case of connecting roads, such as the location of traffic lights, the car's speed generally varies, and sudden changes cause interference in the measurement of the actual speed, but the promising results of the solution elaborate on its effectiveness in crowded roads or junctions where vehicles have variant and non-stable speeds. In addition, our solution has the capability to connect to the traffic police with a friendly and easy-use GUI, which can handle the data and helps the end-users in the smart city to control the camera and record. In future work, we aim to evaluate it using the three different angles (yaw, pitch, roll) along with implementing other deep learning methods and comparing the measured system speed result with the city police control traffic cameras for better calibration and evaluation. It is also planned to check and average the vehicle's speed based on job distribution and a swarm of UAVs.

**Author Contributions:** Conceptualization: A.J.M., A.S.R., L.L., J.W. and A.T.; methodology: A.J.M., A.S.R., L.L. and J.W.; software: A.J.M., A.S.R. and L.L.; validation: A.J.M., A.S.R., L.L. and J.W.; investigation: A.J.M., A.S.R., L.L., J.W. and A.T.; resources: A.J.M., A.S.R. and L.L.; writing—original draft preparation: A.J.M., A.S.R., L.L., A.T. and J.W.; writing-review and editing: A.J.M., A.T., J.W. and M.G. All authors have read and agreed to the published version of the manuscript.

**Funding:** This work was supported by Jiangxi University of Science and Technology, 341000, Ganzhou, P.R China, underfunding numbers: 2021205200100563 (Corresponding author: Jianqing Wu).

**Institutional Review Board Statement:** Not applicable.

**Informed Consent Statement:** Not applicable.

**Data Availability Statement:** The data is available upon the request.

**Conflicts of Interest:** The authors declare no conflict of interest.

## Nomenclature

| | |
|---|---|
| AVG H_FOV | Average Horizontal Field Of View |
| Band | Band values |
| $Coords_A$ | Centre of points A |
| $Coords_B$ | Centre of points B |
| d(x, y) | Distance between x, y |
| $d_{AB}$ | Distance in meters between A & B |
| d/t | Distance over time |
| DL | Distance From Lens |
| DI_L | Diagonal Length |
| D_FOV | Diagonal FOV |
| FOV | Field of View |
| HL | Horizontal Length |
| H_FOV | Horizontal FOV |
| Km/h | Kilometer per hour |
| L1 | Length of base 1 of the trapezoidal image area |
| L2 | Length of base 2 of the trapezoidal image area |
| MT | Median of the trapezoidal image area |
| NormBand | Normalized band |
| OUT | out object |
| PPM | Pixel per meter |
| $P_{AB}$ | Pixel spacing between a & b |
| UAV | Unmanned aerial vehicle |
| VL1 | the Vertical Length of base 1 |
| VL2 | the Vertical Length of base 2 |
| VMT | Vertical Median of Trapezoidal Frame |
| V_FOV | Vertical FOV |
| $\mu_{BAND}$ | The mean of each red, green, and blue band |
| $\Delta t_{AB}$ | Timestamp between a and b |
| $\sigma$ | Scaling factor for normalization |

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
