# Peer review of "A Secure Traffic Police Remote Sensing Approach via a Deep Learning-Based Low-Altitude Vehicle Speed Detector through UAVs in Smart Cites: Algorithm, Implementation and Evaluation"

_futuretransp, doi:10.3390/futuretransp3010012_

Round 1

Reviewer 1 Report

This paper introduces a novel low-altitude vehicle speed detector system using UAVs for remote sensing applications of smart cities, forging to increase traffic safety and security.

The following are some comments regarding the paper:

1- Please rewrite the sentences of first-person statements to third-person (Lines 81-89).

2- There are some errors, especially in the reference part. The authors need to correct them. (Lines 350, 395)

3- There are some grammatical errors in the paper, therefore it is better to check the full text carefully, and avoid grammar or spelling mistakes. Please revise the paper.

4- Why was PI-4 used in this project?

5- what are the rules for internal stickers?

6- Authors are requested to make typesetting strictly according to the paper template.

Author Response

C1: Comments and Suggestions for Authors

This paper introduces a novel low-altitude vehicle speed detector system using UAVs for remote sensing applications of smart cities, forging to increase traffic safety and security.The following are some comments regarding the paper:

  1. Please rewrite the sentences of first-person statements to third-person (Lines 81-89).

Response: Thank you for bringing this point to our attention. The sentences are eddied and added to the manuscript.

Changed sentence:

  1. A more reliable and secure solution to verify vehicle speed using drones instead of current studies is offered.
  2. Introducing the fast speed check system that can accompany the available speed system detection and be used in areas with low access.
  3. To achieve our solution, a low-altitude drone that uses the Mobile Net-SSD algorithm was used to detect the speed of vehicles and has the ability to connect to the traffic police.
  4. To achieve more accurate results, several scenarios were considered in different situations, such as road intersections and environments with sudden speed changes.
  5. The effective cases were tested and calibrated using a drone to increase the solution's speed detection accuracy. The system was run on Raspberry PI4B for its faster-processing speed, and memory capabilities such as capacity and bandwidth benefit the deep learning-based computer vision module to run smoothly.
  6. In addition to our previous contribution, as mentioned above, a mobile camera system was installed on top of the drone, providing various facilities to evaluate our solution at different altitudes and X and Y axes.
  7. In addition, a graphical user interface (GUI) was designed and implemented where we can record the environment status, find the abnormal condition from the speed parameter and generate alarms to the relevant authorities. In addition, through the GUI, we can control the state of the camera, such as its movement.
  8. There are some errors, especially in the reference part. The authors need to correct them.

Response: Thank you for bringing this point to our attention. the cross ref problem sorted by adding the name of the tables The mentioned part are changed and adds to the text. (Lines 350, 395)

Changed sentence:

  • As TABLE 6 shows, the data obtained for the tested FOV is very close to the value declared by the camera manufacturer (53 degrees), which indicates the accuracy of the calculations and experiments
  • 8 shows that the car has a different curve in each direction. Then, to have a better-speed estimation, the results from TABLE 8 were formulated with the help of the curve fitting tool in MATLAB software, and the extracted curve is shown in EQ 10 and 11.
  1. There are some grammatical errors in the paper; therefore, it is better to check the full text carefully and avoid grammar or spelling mistakes. Please revise the paper.

Response: Thank you for your valuable comment; we revised the manuscript's language.

  1. Why was PI-4 used in this project?

Response: Thank you for giving us this opportunity to explain this; Nowadays, Pi processors are used as a reliable tool in many Internet of Things projects due to their abundance and price. Concerning the mentioned parameter in this research, this processor was used. This point is added in the Conclusion and Future Work. The respective sentence is highlighted in the text. The effective cases were tested and calibrated using a drone to increase the solution's speed detection accuracy. The system was run on Raspberry PI4B for its faster-processing speed and memory capabilities such as capacity and bandwidth which benefits the deep learning-based computer vision module, to run smoothly.

Changed sentence: Line 88 -91 :By their small size Raspberry Pi processor and its camera(Pi camera) have been used in various Internet of Things projects and are recognized as a trustable platform [19]. Among the variety of Raspberry Pi types, the PI 4 type has been chosen as the latest available version and high speed in data processing, which is a mandatory requirement for this research.

  1. what are the rules for internal stickers?

Response: Thank you so much for your valuable comment. The changed text added to the manuscript

Changed sentence: Line 129-130.  Another reason for using this is its unique workload-specific hardware acceleration that minimizes data movement.

  1. Authors are requested to make typesetting strictly according to the paper template.

Response: Thank you so much for this valuable comment. The manuscript template was checked, and some parts were revised.

Reviewer 2 Report

A Secure Traffic Police Remote Sensing Approach Via A Deep Learning-Based Low-Altitude Vehicle Speed Detector Through UAVs in 3 Smart Cites: Algorithm, Implementation and Evaluation

Introduction shows weak starting,

It must tell why this problem is taken, what is the motivation behind this.

Like, why such setups of UAV based IoT systems are required at these locations.

Then what are challenges for using such microcontrollers.

Mobile Net- 83 SSD algorithm to detect the speed of vehicles

Why Mobile Net-83 SSD is used above other works, please add justification.

Then Why low attitude vehicles are taken only?

Explain the scenarios in perspective of Joins of roads, speeds of vehicles and types etc.

Then UAV/Drone range for flying was fixed or not? Or what was the scenario?

Table 1 values are distorted?

Figure 6 needs more elaborations

Please explain a bit more.

Line 338 to 339, TABLE 4 shows that the system has a limitation of ±75-degree rotation in the direction of 338 the object inside the frame, so the direction of movement should rotate within ±75….

Why table 4 is not clear here.

 Some of the recent references can be added,

Mishra, D., Khan, A., Tiwari, R., & Upadhay, S. (2018, February). Automated irrigation system-IoT based approach. In 2018 3rd International conference on internet of things: Smart Innovation and Usages (IoT-SIU) (pp. 1-4). IEEE.

Tiwari, Rajeev, Hitesh Kumar Sharma, Shuchi Upadhyay, Shreyansh Sachan, and Ayush Sharma. "Automated Parking System-Cloud and IoT based."

Author Response

Reviewer comment 2: Comments and Suggestions for Authors

A Secure Traffic Police Remote Sensing Approach Via A Deep Learning-Based Low-Altitude Vehicle Speed Detector Through UAVs in 3 Smart Cites: Algorithm, Implementation and Evaluation

  1. Introduction shows weak starting,
  • It must tell why this problem is taken, what is the motivation behind this.
  • Like, why such setups of UAV based IoT systems are required at these locations.
  • Then what are the challenges for using such microcontrollers.

Response: Thank you so much for this precise comment. The reason for choosing this research includes the following: Due to the approximate height of 5 to 3 meters and the current vertical angle of the driving cameras on the roads.1)Examining different and appropriate locations and angles in cases with height and angle restrictions, such as winding roads.2) Examining the performance of quads according to the various development processes and future applications.3) Examining the possibility of using the Internet of Things in traffic cases, especially in cases of speed detection in places with short-term review times, such as schools, which require a review of vehicle speed at a limited and specific time. The main challenge in using microcontrollers is processing speed and low accuracy

Changed sentence: According to the review of the past articles and the investigation of the remaining cases, this research has been carried out with the following objectives. Presenting and reviewing an efficient and up-to-date method based on the Internet of Things to help measure car speed as a quick test. Effective measurement of the height and angle of the speed measurement location, regardless of the type and special capabilities of the camera, especially at the height of fewer than 5 meters, different angles according to the approximate height of 5 to 3 meters, and the current vertical angle of the driving cameras on the roads that are on a high and angled route It is used like winding roads—investigating the performance and challenges of using quads in traffic activities with regard to the various development processes and future applications of the Internet of Things in the use of microcontrollers and the importance of their processing speed and accuracy and examining the possibility of using the Internet of Things in traffic cases, especially in cases of speed detection in places with a short-term investigation of times, such as schools that need to check the speed of a car in a limited and specific time.

  1. Mobile Net- 83 SSD algorithm to detect the speed of vehicles, Why Mobile Net-83 SSD is used above other works, please add justification.

Response: Mobile Net- 83 SSD algorithm is selected based on good accuracy for target detection [Z. Jian, etal 2020]and its ability to detect the object in a Single Shot with the Multibox Detector in real-time processing [ (Y. -C. Chiu , etal 2020)], as well as its small size and speed permit it to implement on embedded system platform[J. Ren et al. 2020]. Along with real-time performance, this model, concerning the experiment aim, can track the object that will tilt the camera on a quad in two axes [C. Gao et al, 2021]. The mentioned text with the following references is added to the text.

  • Jian, Z. Yonghui, Y. Yan, L. Ruonan and W. Xueyao, "MobileNet-SSD with adaptive expansion of receptive field," 2020 IEEE 3rd International Conference of Safe Production and Informatization (IICSPI), 2020, pp. 177-181, doi: 10.1109/IICSPI51290.2020.9332204.
  • -C. Chiu, C. -Y. Tsai, M. -D. Ruan, G. -Y. Shen and T. -T. Lee, "Mobilenet-SSDv2: An Improved Object Detection Model for Embedded Systems," 2020 International Conference on System Science and Engineering (ICSSE), 2020, pp. 1-5, doi: 10.1109/ICSSE50014.2020.9219319.
  • Ren and H. Li, "Implementation of Vehicle and License Plate Detection on Embedded Platform," 2020 12th International Conference on Measuring Technology and Mechatronics Automation (ICMTMA), 2020, pp. 75-79, doi: 10.1109/ICMTMA50254.2020.00025
  • Gao, Y. Zhai and X. Guo, "Visual Object Detection and Tracking System Design based on MobileNet-SSD," 2021 7th International Conference on Computer and Communications (ICCC), 2021, pp. 589-593, doi: 10.1109/ICCC54389.2021.9674450.

   Changed sentence: In this research Mobile Net-83 SSD model was selected due to features such as good accuracy for target detection[20], small size and High speed with real-time processing performance[, the ability to detect the object in one shot with the Multibox detector[22], along with the ability to implement on embedded system platforms. In addition, this model can detect the object[24] that is supposed to tilt the camera UAV in two axes, according to the experimental target[25].

  1. Then Why low attitude vehicles are taken only?

Response: we apricated your comment. The reason for checking the height is to use a camera with fewer pixels and increase the UAV operating hours, as well as not interfere with the wind speed in the performance and control of the quad. This point is written in the conclusion section

   Changed sentence:  The special advantages of this design in low-altitude flights include battery energy consumption, reduced UAV charging speed, longer flight length, and the ability to develop a design to track vehicle speed at dynamic speed points such as intersections, Joins of roads, traffic lights, etc. Another reason to check the vehicle's speed at a low height is to use a camera with fewer pixels and increase the UAV operating hours, as well as not interfere with the wind speed in the performance and control of the quad.

  1. Explain the scenarios in the perspective of Joins of roads, speeds of vehicles and types etc.

 Response: Thank you for the comment. In the case of connecting roads, such as the location of traffic lights, the car's speed generally varies, and sudden changes cause interference in the measurement of the actual speed. this point is mentioned in conclusion section :

Changed sentence:   Even In the case of connecting roads, such as the location of traffic lights, the car's speed generally varies, and sudden changes cause interference in the measurement of the actual speed, but the promising results of the solution elaborate on its effectiveness in crowded roads or junctions where vehicles have variant and non-stable speeds.

  1. Then UAV/Drone range for flying was fixed or not? Or what was the scenario?

Response: Thank you for reminding us about this point. The range of flying UAV height based on the table(6) changed, but to have the scale for measuring the location was fixed.

Changed sentence:  The real vehicle test includes experiments regarding changes in UAV height from 0.7 to 3 m with the fixed location, changing the UAV degree with X-axis from -15 to +15, changing the camera degree with X-axis from -90 to +90, changing the camera degree with Y-axis from -90 to +90.

  1. Table 1 values are distorted?

Response: Thank you for reminding us about this point. As table 2 has the values(Table 1 is pseudocode), we correct the values into 4 digits after the point   

  1. Figure 6 needs more elaborations

Response: Thank you for reminding us about this point. The figure 6 elaborated more bellow the image.   

Changed sentence: Fig. 6 shows the tilting system assembled with UAV and controlled with Rasspeby Pi. As shown in Fig. 6A, using the assembled servo motor can move in ± 90 in X and Y directions Fig. 6B shows that changing the baseline of L1 and L2 can cause the three parallel situations, Fig.6 C: horizontally angled, Fig.6D vertically tilted, which can affect the acquired image by the camera. It should be mentioned that for the Outdoor test, the same assembly without a stand is installed on the top of the UAV.

  1. Please explain a bit more.

Line 338 to 339, TABLE 4 shows that the system has a limitation of ±75-degree rotation in the direction of 338 the object inside the frame, so the direction of movement should rotate within ±75….

Response: Thank you for reminding us about this point. Some text added regarding this section.

Changed sentence: TABLE 4 shows that the system has a limitation of ±75-degree rotation in the direction of the object inside the frame, so the direction of movement should rotate within ±75 degrees. As the result shows, the degree between 75 and 90 in the Y direction with positive and negative values causes the camera disability to record an image, so the system loses its performance to track the object.

  1. Why table 4 is not clear here.

Response: Thank you for reminding us about this point. Table 4 again adds to the text.

  1. Some of the recent references can be added,
  • Mishra, D., Khan, A., Tiwari, R., & Upadhay, S. (2018, February). Automated irrigation system-IoT based approach. In 2018 3rd International conference on internet of things: Smart Innovation and Usages (IoT-SIU)(pp. 1-4). IEEE.
  • Tiwari, Rajeev, Hitesh Kumar Sharma, Shuchi Upadhyay, Shreyansh Sachan, and Ayush Sharma. "Automated Parking System-Cloud and IoT based."

Response: Thank you for your comment the reference added to the text.

Changed sentence: Line – 564 to 568 & 468: This is especially critical for highways and wide roads because vehicles will not always drive along the road edge [31-32].

Ref Section End

  1. 31. Mishra, D., Khan, A., Tiwari, R., & Upadhay, S. (2018, February). Automated irrigation system-IoT based approach. In 2018 3rd International conference on internet of things: Smart Innovation and Usages (IoT-SIU) (pp. 1-4). IEEE.
  2. 32. Tiwari, Rajeev, Hitesh Kumar Sharma, Shuchi Upadhyay, Shreyansh Sachan, and Ayush Sharma. "Automated Parking System-Cloud and IoT based."

Reviewer 3 Report

In this paper, the authors introduce a low-altitude vehicle speed detection system using drones for smart city remote sensing applications, forging to increase traffic safety and security. the proposed solution was implemented on the edge node with the configuration of a PI4B and an Intel Neural Computing 2, together with the PI camera, which is equipped with a Mobile Net-SSD deep learning model for the intelligent detection of vehicles and their speeds. The experimental results proved the superior performance of the proposed low-altitude UAV system compared to current studies to detect and estimate vehicle speeds in highly dynamic situations and at different speeds.

The work is interesting and scientifically sound and contains satisfying results, however there are a few issues:

- The language must be improved in order to correct some errors and to facilitate the reading of this paper.

- The extension of the state of the art by adding the related works section in order to generate related works and to properly situate the contribution among the already existing methods

- The summary of the proposed method in the form of a flowchart is desirable to facilitate understanding of the proposed solution

- Why the use of the PI4B on-board card compared to other on-board cards and a question concerning the quality of the images captured by this camera with the change in weather conditions

Author Response

Reviewer comment 3: Comments and Suggestions for Authors

In this paper, the authors introduce a low-altitude vehicle speed detection system using drones for smart city remote sensing applications, forging to increase traffic safety and security. the proposed solution was implemented on the edge node with the configuration of a PI4B and an Intel Neural Computing 2, together with the PI camera, which is equipped with a Mobile Net-SSD deep learning model for the intelligent detection of vehicles and their speeds. The experimental results proved the superior performance of the proposed low-altitude UAV system compared to current studies to detect and estimate vehicle speeds in highly dynamic situations and at different speeds.

The work is interesting and scientifically sound and contains satisfying results, however, there are a few issues:

1- The language must be improved in order to correct some errors and to facilitate the reading of this paper.

Response: Thank you so much for your valuable comment on the paper language modified in the revised manuscript.

2- The extension of the state of the art by adding the related works section in order to generate related works and to properly situate the contribution among the already existing methods –

Response: Thank you for your suggestion. We rewrite all the required sentences. The authors have revised the text in the main introduction and contribution section.

Changed sentence: Line 56: However, visual/object tracking to locate, detect, and define objects [10] via UAV images is still challenging, even with state-of-the-art deep learning models (RESNET, Mobilenet, Efficient Net, etc. ) in the field of computer vision.

Changed sentence: Line 80: According to the review of the past articles and the investigation of the remaining cases, this research has been carried out with the following objectives. Presenting and reviewing an efficient and up-to-date method based on the Internet of Things to help measure car speed as a quick test. Effective measurement of the height and angle of the speed measurement location, regardless of the type and special capabilities of the camera, especially at the height of fewer than 5 meters, different angles according to the approximate height of 5 to 3 meters, and the current vertical angle of the driving cameras on the roads that are on a high and angled route It is used like winding roads—investigating the performance and challenges of using quads in traffic activities with regard to the various development processes and future applications of the Internet of Things in the use of microcontrollers and the importance of their processing speed and accuracy and examining the possibility of using the Internet of Things in traffic cases, especially in cases of speed detection in places with a short-term investigation of times, such as schools that need to check the speed of a car in a limited and specific time.This paper makes the following contributions:

  1. A more reliable and secure solution to verify vehicle speed using drones instead of current studies is offered.
  2. Introducing the fast speed check system that can accompany the available speed system detection and be used in areas with low access.
  3. To achieve our solution, a low-altitude drone that uses the Mobile Net-SSD algorithm was used to detect the speed of vehicles and has the ability to connect to the traffic police.
  4. To achieve more accurate results, several scenarios were considered in different situations, such as road intersections and environments with sudden speed changes.
  5. The effective cases were tested and calibrated using a drone to increase the solution's speed detection accuracy. The system was run on Raspberry PI4B for its faster-processing speed, and memory capabilities such as capacity and bandwidth benefit the deep learning-based computer vision module to run smoothly.
  6. In addition to our previous contribution, as mentioned above, a mobile camera system was installed on top of the drone, providing various facilities to evaluate our solution at different altitudes and X and Y axes.
  7. In addition, a graphical user interface (GUI) was designed and implemented where we can record the environment status, find the abnormal condition from the speed parameter and generate alarms to the relevant authorities. In addition, through the GUI, we can control the state of the camera, such as its movement.

3- The summary of the proposed method in the form of a flowchart is desirable to facilitate understanding of the proposed solution

Response: Thank you for your valuable comment. We have rewritten the process in pseudocode to clarify the method. Besides,to make the article clearer, this part has been moved to section 2.3  part. Also, a flow chart has been designed, but as it is similar, the table 1 code is not added to the main text, but if the respected referee insists, it can replace with Table 1.

2.3 Vehicle Detection and Tracking ;

The proposed system follows the described process in Table 1. This table shows some steps, initializing the camera for live stream, DNN (SSD-mobile net) calling to perform as the vehicles detection section, and speed calculation.

Table 1. The pseudo-code for the whole vehicle speed estimation process.

Start

1

2

Initialize program

   Start Video stream: Get the current frame

3

Call OpenCV DNN: Get detected vehicles

4

Loop through detections:

5

     Vehicle detection confidence estimation

6

If (confidence > set min confidence):

7

8

9

10

11

Assign an index to each detected vehicle.

Define a bounding box around the vehicle.

Compute the x and y coordinates of a box.

Set the tracker to the box.

Set start x, y coordinates.

12

Else:

13

14

15

16

For each vehicle tracker:

Get updated position.

Update position with the new end x, y coordinates.

Add the tracked object to the list

17

18

19

20

21

22

Speed Calculate: 

Meters per pixel.

Distance in pixels.

Distance in meters per moving point.

Average distance.

Speed in km.

End Program

 Table 1 depicts the whole process of the Vehicle detection system, where the program takes in each frame of Livestream footage and runs it through a pre-trained deep learning model. After initialization and getting the object frame called, the OpenCV DNN converts the image blob and gets the detection by confidence evaluation. The model contains classification methods for all vehicles, such as cars, buses, trucks, etc. The program then puts a bounding box around the detected vehicle. It tracks it as the vehicle moves across consecutive frames and calculates and declares the speed using the formulas explained in this section.

4- Why the use of the PI4B onboard card compared to other onboard cards and a question concerning the quality of the images captured by this camera with the change in weather conditions

Response: Thank you for giving us this opportunity to explain this; Nowadays, Pi processors are used as a reliable tool in many Internet of Things projects due to their abundance and price. Concerning the mentioned parameter in this research, this processor was used. The weather conditions can affect any camera, and mainly under the conditions like heavy rain, there are some restrictions for UAVs,  but the small size of the camera was the main aim of this selection. This point is added in the Conclusion and Future Work. The respective sentence is highlighted in the text.

Changed sentence: By their small size Raspberry Pi processor and its camera(Pi camera) have been used in various Internet of Things projects and are recognized as a trustable platform [19]. Among the variety of Raspberry Pi types, the PI 4 type has been chosen as the latest available version and high speed in data processing, which is a mandatory requirement for this research.

Reviewer 4 Report

The article is interesting and contains a lot of important information. It is certainly worth publishing, but it needs some improvements and clarifications.

1. What are the main advantages of using the Raspberry PI Field of View (FOV) as opposed to other devices of this type?

2 Are there grounds to question the results of the measurements in this experiment?

3. What is the aforementioned improvement of the methods of non-intelligent cities, please give some numerical values, percentages?

Author Response

Reviewer comment 4: Comments and Suggestions for Authors

The article is interesting and contains a lot of important information. It is certainly worth publishing, but it needs some improvements and clarifications.

  1. What are the main advantages of using the Raspberry PI Field of View (FOV) as opposed to other devices of this type?

Response: Thank you for giving us this opportunity to explain this; Nowadays, researchers have accepted the necessity of using small components as an integral part of designs. In this design also, the PI camera with respect to its weight and dimensions has been selected and used. Besides as mentioned in Gay, W. (2018) and other online resources the usage of the graphics processing capability of the Broadcom CPU is one of the major advantages of this camera over a USB camera that makes the pi camera unique.

Gay, W. (2018). Pi Camera. In: Advanced Raspberry Pi. Apress, Berkeley, CA.https://doi.org/10.1007/978-1-4842-3948-3_26

This text is added in lines 158-161;

2 Are there grounds to question the results of the measurements in this experiment?

Response: Thank you for your commentï¼› The measured speed values in this research are reported in Tables 8 and 10, although it is important to note that based on the available literature, speed detection at the position of intersections and areas where the vehicle suddenly has a dynamic speed has not been reported and the proposed designed system can be used as a benchmark test. In addition, during the experiment, we calibrated the measured value of the system with the actual speed of the car once and compared our result with a mobile phone application named Speed GUN, but as a limitation, it can be said that it is not possible to compare the results With the cameras of the Chinese traffic police in the testing place, due to security problems, which is mentioned as in the conclusion section.

This text is added in lines 555-560;

  1. What is the aforementioned improvement of the methods of non-intelligent cities, please give some numerical values, percentages?

Response: Thank you for your comment, even the clear percentage for method improvement in non-intelligent cities is not reported but The mentioned improvement and utilization of modern technologies in making cities smart specially in the transportation department increases various parameters such as economy, environment, and infrastructure from 10% to 30% [37]. And the data (image, video, graphics etc) from these applications can help find behaviours and predictions using Big Data Learning and Discovery techniques [38]. 

This text is added in lines 533-538;

The authors appreciate the dear reviewer for the careful consideration and useful comments on the paper. The authors especially thank the respected reviewer for the valuable time he/she has devoted to reviewing the manuscript. Based on these valuable comments, the article has been revised. The authors hope that the revised paper will meet the expectations of the respected reviewer. We wish you all the best.

Warm regards;

Reviewer 5 Report

The work of the authors is very interesting and innovative. Indeed, with the development of technology, there is a need for a thorough study of urban traffic, especially in large urban agglomerations. In such cities, it is difficult to accurately measure vehicle speed and the number of vehicles passing in front of a given intersection.

In the abstract, the authors did not specify the main purpose of the article. The main purpose of the article appears only in the introduction (77-80). Please specify the main purpose of the research. Because the goal based on the measurement of the vehicle speed is very general and it should be clarified to determine the impact of selected parameters on the read vehicle speed.

The introduction of 80 to 122 should be in the methodology. In this part, the authors describe in detail their contribution to the development of the scientific discipline. This should be emphasized in the methodology chapter.

Figure 1 should be placed before section 2.1. In addition, authors must pay attention to the form of referring to figures or tables. If we give a reference in parentheses ( ), we use an abbreviated reference (Fig. ). And if the reference is in the text, the entire name of the Figure should be given.

The diagram in Table 1 is not very clear. Maybe the authors would consider presenting this scheme using a graphic drawing??

The authors presented the method of measuring the speed of vehicles and detecting vehicles. This part of the work is very interesting and innovative. However, the authors should focus on a few additional aspects. It was necessary to check whether the camera used does not distort the real image. In addition, the authors show large differences between the speed of the moving vehicle and the recorded speed. The difference up to 13 km/h is very large, in some EU countries exceeding the speed limit by several km/h is punishable by losing your driving license. Devices designed to measure the speed of vehicles have a margin of error of up to 3 km/h, such devices are acceptable, so this difference should be reduced. It is necessary to specify in what conditions the camera should work in order to obtain the most optimal result. The authors checked the height from 0.7 to 3 m and obtained greater accuracy at 3 m, perhaps it would be even better at 3.5 m and maybe 3.5 m would already cause deterioration. The authors must clarify why they chose this range and whether they conducted tests at other altitudes.

In addition, the device should be compared with another device used to measure the speed of vehicles or count vehicles. The article lacks information on what vehicle was tested and how they maintained a constant speed.

Author Response

Reviewer comment 5: Comments and Suggestions for Authors

The work of the authors is very interesting and innovative. Indeed, with the development of technology, there is a need for a thorough study of urban traffic, especially in large urban agglomerations. In such cities, it is difficult to accurately measure vehicle speed and the number of vehicles passing in front of a given intersection.

  1. In the abstract, the authors did not specify the main purpose of the article. The main purpose of the article appears only in the introduction (77-80). Please specify the main purpose of the research. Because the goal based on the measurement of the vehicle speed is very general and it should be clarified to determine the impact of selected parameters on the read vehicle speed.

Response: Thank you so much for adding your valuable comment on the mentioned point in the revised manuscript.

This text is added in lines 36-38;

  1. The introduction of 80 to 122 should be in the methodology. In this part, the authors describe in detail their contribution to the development of the scientific discipline. This should be emphasized in the methodology chapter.

Response: Thank you so much for this precise comment. The related text are move to the methodology part the text changed.

This text is added in lines 116-128;

  1. Figure 1 should be placed before section 2.1.

Response: we apricated your comment. the figure position changed.

  1. In addition, authors must pay attention to the form of referring to figures or tables. If we give a reference in parentheses ( ), we use an abbreviated reference (Fig. ). And if the reference is in the text, the entire name of the Figure should be given.

Response: we apricated your comment the change has been done in the manuscript.

  1. The diagram in Table 1 is not very clear. Maybe the authors would consider presenting this scheme using a graphic drawing??

Response: Thank you for your valuable comment. a flow chart has been designed and replaced with the pseudo-code along with some text.

This text is added in lines 201-216; ,

  1. The authors presented the method of measuring the speed of vehicles and detecting vehicles. This part of the work is very interesting and innovative. However, the authors should focus on a few additional aspects. It was necessary to check whether the camera used does not distort the real image. In addition, the authors show large differences between the speed of the moving vehicle and the recorded speed. The difference up to 13 km/h is very large, in some EU countries exceeding the speed limit by several km/h is punishable by losing your driving license. Devices designed to measure the speed of vehicles have a margin of error of up to 3 km/h, such devices are acceptable, so this difference should be reduced. It is necessary to specify in what conditions the camera should work in order to obtain the most optimal result.

Response: Thank you for reminding us about this point. The measured speed values in this research are reported in Tables8,9 and 10, although it is important to note that based on the available literature, speed detection at the position of intersections and areas where the vehicle suddenly has a dynamic speed has not been reported and the proposed designed system can be used as a benchmark test. As mentioned in the experiment process the best distance with respect to the camera FOV with the experimented Hight is 15 meters and even the measured speeds are calibrated with real car speed but it's better that rely on the speed value of the car which is positioned in the same side of UAV  this point is mentioned in the conclusion section.

This text is added in lines 518-529; ,

  1. The authors checked the height from 0.7 to 3 m and obtained greater accuracy at 3 m, perhaps it would be even better at 3.5 m and maybe 3.5 m would already cause deterioration. The authors must clarify why they chose this range and whether they conducted tests at other altitudes.

Response: Thank you for reminding us about this point. As mentioned by the police rule in china and other security issues more than 3 meters was not affordable in the practical test. also, the height of 4 meters to 5 meters is the height that most of the current traffic cameras are installed. In addition, as one of the questions addressed in this research, is to show the ability to measure the speed at low altitudes.

This text is added in lines 471-472

  1. In addition, the device should be compared with another device used to measure the speed of vehicles or count vehicles. The article lacks information on what vehicle was tested and how they maintained a constant speed.

Response: Thank you for reminding us about this point. during the experiment, we calibrated the measured value of the system with the actual speed of the car once and compared our result with a mobile phone application named Speed GUN, but as a limitation, it can be said that it is not possible to compare the results. With the cameras of the Chinese traffic police in the testing place, he pointed out that due to security problems, which is mentioned as a limitation in the conclusion section. In the calibration test, the UAV   was positioned in the fixed-point test then In order to have a uniform speed on a straight road with a speed limit of 80 km per hour, a car with a driver and one of the researchers as an assistant driver was used. Before the vehicle starts to move the driver has information about the  target speed then the driver brings the car's speed to the desired speed at the UAV location. Then, when passing the quad position, along with checking Vehicle identification the UAV speed reading and the Vehicle speed from odometer  read and record  by the driver's assistant.

This text is added in lines 430-442.

The authors appreciate the dear reviewer for the careful consideration and useful comments on the paper. The authors especially thank the respected reviewer for the valuable time he/she has devoted to reviewing the manuscript. Based on these valuable comments, the article has been revised. The authors hope that the revised paper will meet the expectations of the respected reviewer. We wish you all the best.

Warm regards;

Reviewer 6 Report

This paper introduces a novel low-altitude vehicle speed detector system using UAVs for remote sensing applications of smart cities, forging to increase traffic safety and security.

The topic of the paper is interesting, however, there are few minors issues.  Detailed comments are given as:

lThe motivation of the paper should be improved.

lAuthors have used several notations, I recommend to add a table that contains all the notations. Also check the whole manuscript for typos and grammar errors.

lLanguage of the paper should be improved.

lThe future work “In future work, we aim to evaluate it using the three angles (yaw, pitch, roll) indifferent” looks incomplete, it is suggested to add more details about future work.

The current literature in the manuscript does not reflect the related work clearly, the authors should add some recently highly quality published papers, e.g., DOI: 10.1007/s00500-021-06465-y;  DOI: 10.3390/electronics11213551; DOI: 10.1109/ACCESS.2021.3111318; DOI: 10.1007/s11235-022-00930-1.

Author Response

Reviewer comment 6: Comments and Suggestions for Authors

This paper introduces a novel low-altitude vehicle speed detector system using UAVs for remote sensing applications of smart cities, forging to increase traffic safety and security. The topic of the paper is interesting, however, there are few minor issues.  Detailed comments are given as:

  1. The motivation of the paper should be improved.

Response: Thank you for bringing this point to our attention. The improved text inserted in section 2.

This text is added in lines 81-111.

  1. Authors have used several notations, I recommend to add a table that contains all the notations. 

Response: This part was added at the end before the reference section and sorted based on alphabetic order .

  1. Also check the whole manuscript for typos and grammar errors. Language of the paper should be improved.

Response: Thank you so much for your valuable comment. The changed text added to the manuscript

  1. The future work “In future work, we aim to evaluate it using the three angles (yaw, pitch, roll) indifferent” looks incomplete, it is suggested to add more details about future work.

Response: Thank you so much for this valuable comment. the text revised.

This text is added in lines 551-561.

  1. The current literature in the manuscript does not reflect the related work clearly, the authors should add some recently highly quality published papers, e.g.,DOI: 10.1007/s00500-021-06465-y;  DOI: 10.3390/electronics11213551;DOI: 10.1109/ACCESS.2021.3111318; DOI: 10.1007/s11235-022-00930-1.

Response: Thank you so much for your valuable comment. All mentioned papers are added as reference number [35-38]

The authors appreciate the dear reviewer for the careful consideration and useful comments on the paper. The authors especially thank the respected reviewer for the valuable time he/she has devoted to reviewing the manuscript. Based on these valuable comments, the article has been revised. The authors hope that the revised paper will meet the expectations of the respected reviewer. We wish you all the best.

Warm regards;